# Genomics and Epigenomics in the Molecular Biology of Melanoma—A Prerequisite for Biomarkers Studies

**DOI:** 10.3390/ijms24010716

**Published:** 2022-12-31

**Authors:** Daniela Luminita Zob, Iolanda Augustin, Lavinia Caba, Monica-Cristina Panzaru, Setalia Popa, Alina Delia Popa, Laura Florea, Eusebiu Vlad Gorduza

**Affiliations:** 1Department of Medical Oncology, AI. Trestioreanu Institute of Oncology, 022328 Bucharest, Romania; 2Department of Medical Genetics, Faculty of Medicine, “Grigore T. Popa” University of Medicine and Pharmacy, 16 University Street, 700115 Iasi, Romania; 3Nursing Department, Faculty of Medicine, “Grigore T. Popa” University of Medicine and Pharmacy, 16 University Street, 700115 Iasi, Romania; 4Department of Nephrology-Internal Medicine, Faculty of Medicine, “Grigore T. Popa” University of Medicine and Pharmacy, 16 University Street, 700115 Iasi, Romania

**Keywords:** cutaneous melanoma, genomics, epigenomics, biomarkers

## Abstract

Melanoma is a common and aggressive tumor originating from melanocytes. The increasing incidence of cutaneous melanoma in recent last decades highlights the need for predictive biomarkers studies. Melanoma development is a complex process, involving the interplay of genetic, epigenetic, and environmental factors. Genetic aberrations include *BRAF*, *NRAS*, *NF1*, *MAP2K1/MAP2K2*, *KIT*, *GNAQ*, *GNA11*, *CDKN2A*, *TERT* mutations, and translocations of kinases. Epigenetic alterations involve microRNAs, non-coding RNAs, histones modifications, and abnormal DNA methylations. Genetic aberrations and epigenetic marks are important as biomarkers for the diagnosis, prognosis, and prediction of disease recurrence, and for therapeutic targets. This review summarizes our current knowledge of the genomic and epigenetic changes in melanoma and discusses the latest scientific information.

## 1. Introduction

Melanoma is a malignant tumor originating from melanocytes, the cells specialized in producing the melanin pigment. Melanocytes emerge from the neural crest, an embryonic structure consisting of migratory pluripotent cells from which several different cell types originate [1]. In the maturation process, melanocyte progenitors migrate, differentiate, and colonize the skin, hair follicles, uvea, and mucous membranes throughout the body. Accordingly, melanoma can arise in any of these sites, leading to a genetically, histologically, phenotypically, and clinically heterogeneous disease. 

In Caucasian populations, the most common type of melanoma is cutaneous melanoma (CM). In recent decades, a continuous increase of CM frequency rates has been observed in Caucasian populations worldwide, making CM the cancer with the most rapidly increasing occurrence [2]. Melanoma is dangerous because it poses a greater risk for metastasizing than other skin cancers. [3]. 

Cutaneous melanoma comprises four main subtypes: superficial spreading melanoma, lentigo melanoma, acral lentiginous melanoma, and nodular melanoma. Superficial spreading melanoma accounts for about 70% of melanoma cases. It is more frequent in fairer-skinned patients, the locations are the trunk and legs, and the excessive UV exposure is a risk factor [4]. Lentigo melanoma targets older patients with sun- damaged skin and the affected skin areas are the face, ears, and neck [5].

Acral lentiginous melanoma is typically found in individuals with darker skin. Clinical presentation is a dark spot on the sole or palms. Subungual melanoma represents a subcategory of acral lentiginous melanoma. Clinically, the lesions are localized under the nail beds and appear as dark vertical streaks. Sun exposure is not a risk factor for this rare form of melanoma [5]. Nodular melanoma accounts for 15% of cases. It is the most aggressive subtype of melanoma and usually affects fair-skinned people over 65. All races could be affected and clinical appearance is a firm lump/node that arises on the skin’s surface [6].

The American Joint Committee on Cancer (AJCC) proposed scoring parameters for clinical classification of melanoma. This classification takes into consideration the thickness of the primary tumor (T) and its subcategory, the presence of ulceration, the regional lymph node involvement (N), and the presence of distant metastases (M). In patients with distant metastases, there are two categories according to serum lactate dehydrogenase (LDH) levels: 0—with low LDH level and 1—with elevated LDH level. However, TNM classification is the most important prognostic factor, and mitotic rate, age, and gender are also valuable prognostic factors.

Well-known clinicopathologic characteristics define traditional parameters for melanoma staging and prognosis. Regarding primary tumors, tumor thickness and ulceration are strong predictors of survival. The degree of vascular invasion also significantly influences outcome, but only in the set of thin melanomas (<1 mm). In the N category, three independent elements have prognostic value: the number of metastatic nodes, whether nodal metastases were clinically occult or apparent, and the presence or absence of musculoskeletal injury. Moreover, patients with clinically palpable nodes have a shorter survival than patients with non-palpable disease. The M status, as well as the localization and the number of metastatic sites, is also of great importance for determining the prognosis of a patient [7].

LDH is important for the prediction of survival in advanced stages of melanoma. An elevated serum level of this marker is independent and significant for such a predictor [8].

CM development is a complex multi-factorial process, arising from multiple etiologic pathways and involving the interplay of genetic and environmental risk factors. The two main causes of melanoma are extrinsic UV exposure and genetic predisposition (family history and phenotypic traits carrying a strong genetic component, including hair and eye color and the number of common and atypical melanocytic nevi) [9,10].

Recent decades have brought about a major impact on improving survival in melanoma patients. Immunotherapy (independently of genomic mutations) and the combination of BRAF and MEK inhibitors (in patients with BRAF V600 mutations who have a poorer prognosis) improved overall survival [11,12]. At the same time, new molecular profiles were identified. However, mortality remains high in conditions where more than half of the patients with metastases die within five years [13].

The research studies in recent years in the field of oncology, especially using high-throughput technologies, constitute a premise for personalized medicine and precision medicine at the same time. Thus, the determination of gene expression profile, disease-related genes, and gene signature allows for a better grouping of patients for the purpose of targeted individualized treatment. For example, some studies demonstrated the association between mRNA-signatures and prognosis in melanoma patients [14,15].

## 2. Genomics in Melanoma and Implications for Biomarkers

Genetic aberrations, concurrently with epigenetic characteristics and the tumor microenvironment (TME), lead to tumor progression that eventually results in uncontrolled cell proliferation, an escape of immune annihilation, and induces a metastatic potential. Discovery of the genomic modifications of melanocytic tumors has remarkably enhanced our understanding of pathogenesis. Thus, we established correlations between the genetic abnormalities and the clinical and pathologic characteristics. The uncontrolled cell proliferation of melanocytes depends on the mutations in some oncogenes, such as *MC1R*, *CDK4*, *BRAF*, *CCND1*, *RAS*, *NRAS*, *c-KIT*, *GNAQ*, and *GNA11* or in tumor suppressor genes, such as, *TP53*, *BCORL1*, *PPP2R3B*, *RASA2*, *PTEN*, and *CDKN2A* [16]. The main signalling pathways of melanoma are: MAPK (mitogen-activated protein kinase), AKT (protein kinase B) pathway (PI3K/PTEN/AKT), cell-cycle regulation pathway, pigmentation-related pathway (MITF signalling pathways), and p53 pathway [17].

The progression to melanoma involves a minimum of three key mutations, but the number of genetic abnormalities is high in melanoma. Sometimes, the occurrence of mutations is so fast that a precursor lesion exists only for a short time, which is not enough to detect them [18,19,20,21].

Genomic classification of cutaneous melanoma includes four main categories: *BRAF*-mutated melanoma (50% of all cases); *RAS*-mutated melanoma (25%; *NRAS* is the most commonly mutated); *NF1*-mutated melanoma (10–15%); and the triple-wild-type melanoma (10%; rare variants in *KIT*, *GNAQ*, *GNA11,* or translocations of kinases) [21].

The MAPK pathway plays a major role in the regulation of cell proliferation, cell survival, invasion, the angiogenesis process, and metastasis. Mutant Ras proteins play a role in oncogenesis by activation of the downstream cascade without the stimulation of the respective upstream pathway. Furthermore, it has been suggested that the activation of BRAF, NRAS, and PI3K can occur in various stages of melanoma development [22].

Up to 30% of melanomas are linked to melanocytic precursor lesions, including nevi or intermediate melanocytic tumors [23,24]. It is believed that the initial mutations affect the MAPK pathway, leading to the formation of nevi. When no other genomic aberration is present, melanocytes with a MAPK pathway-activating mutation allow for an oncogene-induced senescence. If benign or intermediate melanocytic tumors develop into further genetic aberrations in cancer-related genes, they are liable to progress and acquire metastatic prospects [25].

The genes from MAPK pathways involved in melanoma pathogenesis are summarized in Table 1.

### 2.1. BRAF

*BRAF* is a *RAF* gene. The connection in the signaling pathway is with the small G protein RAS and through MEK1/2 activate ERK1/ERK2 [3].

*BRAF* mutations represent a frequent event in oncogenesis, mainly in melanoma [28]. The most common mutation (75%) is V600E, when the valine is replaced with glutamic acid in position 600.

*BRAF* mutations appear in the first stages of melanocytic tumors [29], whereas individual studies indicated that *BRAF* mutations have an unfavorable prognosis in patients with melanoma [30,31], which has not been definitively shown in other studies [32].

*BRAF* mutation is implicated in melanoma progression, sustained angiogenesis, tissue invasion, and metastasis, as well as the evasion of the immune response. Multiple studies assessed the prognostic importance of *BRAF* mutations but their role remains controversial. Most studies found a positive association between *BRAF* mutation and poor clinical outcome. The analysis performed in the phase III Keynote 054 trial showed that there are differences between the relapse-free survival (RFS) in patients with BRAF mutations compared to those without BRAF mutations only in the group treated with a placebo. There is no difference in patients treated with pembrolizumab [33]. Other studies suggested an opposite correlation between BRAF mutation and patient outcome prediction. For example, Tas and Ertuk investigated the prognostic value of BRAF V600E mutation in 151 stage III patients. The mutation was present in 51% of patients and was associated with better overall survival (OS) and longer disease-free survival [34].

A recent study indicates that extracellular vesicles (EVs) could be a promising source of mutant DNA for *BRAF* mutation status for evaluating BRAF therapy [35].

### 2.2. NRAS

*NRAS* is a member of the *RAS* family of oncogenes and was originally found in oncogenic viruses [36]. *NRAS* was found to be mutated in melanoma cell lines [37]. *NRAS* mutations emerge in about 20% of melanomas and are evenly spread among cutaneous, acral, and mucosal melanomas [20,21]. In opposition, variants in *HRAS* (HRas proto-oncogene, GTPase) and *KRAS* (KRAS proto-oncogene, GTPase) are less common: 2% and 3%, respectively, of all melanoma [26]. These mutations are recurrent and they are mutually exclusive with other *RAS* gene mutations [38].

Mutations in *RAS* genes generally concern codons 12, 13, and 61. Whereas most aberrations in *NRAS* involve the glutamine on position 61 (Q61), mutations in *KRAS* and *HRAS* are usually found at the glycine 12 and 13 (G12, G13). All three hotspot mutations (Q61, G12, and G13) contribute to GTPase inactivation, resulting in a constitutively functional GTP-bound protein. Activation of the RAS protein signals is generated via multiple oncogenic downstream paths, including activation of the MAPK signaling pathway by the RAS/RAF/MEK/ERK signaling cascade and activation of the PI3K/AKT pathway through PI3K (phosphatidylinositol 3OH-kinase) phosphorylation [37].

### 2.3. NF1

*NF1* (neurofibromin 1) is a tumor suppressor gene [39]. It intervenes as an oncogene in different cancers and it was seen to be non-functional in multiple human malignancies, including lung cancers, neuroblastomas, and glioblastomas [40,41]. *NF1* encodes the protein neurofibromin, which negatively regulates RAS by hydrolysis of RAS-bound GTP to GDP [42]. Functional inactivation of *NF1* contributes to the activation of *RAS* and its downstream signaling pathways, including the MAPK, PI3K/AKT, and mTOR pathways [43].

It has been observed that *NF1* inactivating mutations occur in melanomas without *BRAF* and *NRAS* mutations and cause the activation of the MAPK pathway. *NF1* mutations are typically inactivating, often truncating mutations or losses, and there are no mutation hotspots. Usually, melanomas occurring in heavily sun-damaged skin or desmoplastic melanomas show a significantly higher rate of *NF1* mutations [44,45,46]. *NF1* mutations are more frequent in acral and mucosal melanomas [47,48].

Studies have pointed out that *NF1* could be associated with resistance to BRAF and MEK inhibitors [49,50]. Nevertheless, *NF1*-mutated melanomas have been associated with tumors with a high tumor mutation burden (TMB) that respond well to immunotherapy [51,52]. These discoveries demonstrate that the *NF1* mutational status could be important for therapeutic decision making.

### 2.4. MAP2K1/MAP2K2 (MEK1/MEK2)

*MAP2K1* and *MAP2K2* encode the protein kinases MEK1 and MEK2. They function specifically in the MAPK/ERK cascade and determine the activation of MAPK3/ERK1 and MAPK1/ERK2 and further transduction of the signal within the MAPK/ERK cascade [53]. *MAP2K1/MAP2K2* mutations have been associated with resistance to RAF and MEK inhibitors [54,55].

### 2.5. KIT

The *KIT* gene encodes the mast/stem cell growth factor receptor Kit [56]. *KIT* gene modifications (mutations, copy number variations) are mutually exclusive to *BRAF*, *NRAS*, and *NF1* mutations. The most involved exons are 11 and 13. The L576P and K642E mutations have a positive response to KIT inhibitors, but this is a temporary answer [57,58,59,60].

Translocations represent another mechanism that activates the MAPK and other oncogenic pathways. They are more frequent in tumors with spitzoid morphology. In these cases, fusion gene results in *ALK*, *ROS1*, *RET*, *BRAF*, *NTRK1*, *NTRK3*, and *MET* genes [61,62]. These translocations were determined as mutually exclusive of each other and other MAPK-activating mutations, sustaining the pinpointed translocations as being critical driver events.

*BRAF* translocations occur in melanomas [63]. In this way, the MAPK and PI3K signaling pathways are activated [61].

### 2.6. CDKN2A

*CDKN2A* was first associated with familial melanoma predisposition [64]. Latter studies recognized *CDKN2A* as an essential gene in this locus, responsible for controlling the cell cycle [65]. *CDKN2A* is the most often concerned tumor suppressor gene in sporadic melanoma [66]. *CDKN2A* losses, frequently biallelic, are seen in 50–80% of sporadic melanoma [20,67]. The most common abnormalities are inactivating mutations and promoter methylation [68].

The *CDKN2A* gene encodes for p16 and p14ARF. p16 is important in cell cycle control. It acts on CDK 4/6 and blocks the phosphorylation of the retinoblastoma protein (Rb) [69,70]. Rb phosphorylation releases the E2F transcription factor with cell cycle progression from G1 to the S phase [71]. p14ARF inhibits the ubiquitin ligase MDM2, which targets proteasomal degradation of TP53 [70].

### 2.7. PTEN

*PTEN* is a tumor suppressor gene with inactivating action in the PI3K signaling pathway. The PTEN mainly determines the inhibition of the AKT pathway.

*PTEN* has been reported to be mutated or nonexistent in up to 70% of melanomas [20]. Epigenetic silencing of *PTEN* may likewise play a role in melanoma [72]. *PTEN* loss is more often found in *BRAF*-mutant than in *NRAS*-mutant melanoma [67], a finding compatible with PI3K/AKT pathway activation by *NRAS* but not *BRAF* mutations, which need an additional hit. *PTEN* inactivation has been linked to resistance to BRAF inhibitors [73] and immunotherapy [74] in melanoma patients.

### 2.8. TERT Promoter Mutations

Mutations in the telomerase reverse transcriptase (*TERT*) promoter melanomas are activatied in melanoma [75]. The effects are an increased gene expression, and cell proliferation without senescence or apoptosis [76]. *TERT* promoter mutations were demonstrated to promote tumorigenesis in two phases, mainly by repairing the shortest telomeres [77]. These mutations are markers of aggressive behavior and also of inferior prognosis [78].

### 2.9. TYRP1

The microarray analysis of melanoma metastasis conducted by Journe found that the gene *TYRP1* was associated with shorter survival. *TYRP1* expression in the validation group demonstrated a powerful correlation between TYRP1 protein level and distant metastasis-free survival and OS. This discovery indicates *TYRP1* as a possible prognostic marker for stage III melanoma patients [79].

Even though nowadays we know that *TYRP1* plays an important role, the link with patient survival and how its expression affects cell behaviour is still unclear [14].

### 2.10. ctDNA and CTC

In a recent systematic review, Gandini et al. summarized the studies about the link between ctDNA and survival in over 2000 stage III and IV melanoma patients. They observed that detectable ctDNA before treatment and during follow-up correlated to poorer progression-free survival (PFS) and OS, with no differences by tumour stage or systemic regimen. Even though ctDNA has a high potential as a prognostic biomarker, the standardization of a methodology is necessary before introducing liquid biopsy in clinical practice [80].

### 2.11. Common Variants

A recent meta-analysis genome-wide association study identified 54 significant loci for melanoma and 68 independent SNPs (Single nucleotide polymorphisms) in the meantime. These variants concern genes or loci located in the vicinity of genes involved in pathogenic pathways of melanoma such as DNA repair and telomere length, differentiation of melanocyte and cell adhesion, and immunity [81].

The most common variants associated with melanoma that present a DNA repair pathway modification are rs78378222 and rs161548 at the TP53 locus. In this case, the absence of the normal response of TP53 to cellular stress determines the cellular hyperproliferation of mutant cells [81]. The loci and the SNPs associated with telomere maintenance correspond to the following genes: *POT1* (rs4731207), *TERC* (rs3950296), *RTEL1* (rs143190905), *MPHOSPH6* (rs2967383), *STN1* (rs7902587), *CCND1* (rs4354713), *ATM* (rs1801516), and *PARP1* (rs2695237) [81].

SNPs identified in following genes (or genetic regions located in their proximity) are implied in melanocyte development and differentiation pathway: *FOXD3* (rs670318), *NOTCH2* (rs2793830), *MITF* ( rs149617956), *NOTCH1,* and *SOX10* [81].

E-cadherin encoded by the *CDH1* gene (cadherin 1) plays a major role in the adhesion between melanocytes and keratinocytes. E-cadherin expression is lost in the stage of melanoma progression. SNP rs4420522 in the *CDH1* gene is a risk allele [81].

Common variants in immunity genes associated with melanoma susceptibility have been identified, for example, rs28986343 at the HLA locus and association between rs408825 and *MX2* gene. On the other hand, rs1126809 in the *TYR* gene or rs6059655 in the *ASIP* gene are protective effects [81].

### 2.12. Copy Number Variations (CNV)

Numerous numerical and structural chromosomal aberrations, unlike benign proliferations that do not have such changes, were discovered in malignant melanomas [82]. Chromosomes most frequently involved are 1,6,7,9,10, and 11. These chromosomal regions contain genes [*GNAQ* (9q21.2), *BRAF* (7q34), *PTEN* (10q23.31), *CCND1* (11q13.3), *RREB1* (6p24.3), *MYB* (6q23.3), and *CDKN2A* (9p21.3)] involved in the MAPK pathway, which is consistent with the involvement of this pathway in the pathogenesis of melanoma [82,83].

Copy number variations can be generated by loss or gain of genetic material and these correlate with the stage of tumorigenesis, patient age, and histological type [82,83].

The most frequent anomalies are losses of chromosome 9 (81% of tumours) and chromosome 10 (63% of tumours), with both anomalies appearing in the early stage of tumorigenesis. At the level of chromosome 9, usually, region 9p21 where the *CDKN2A* gene is located, which encodes cyclin-dependent kinase inhibitor 2A, is concerned. It interacts with CDK4 (cyclin-dependent kinase 4) and CDK6 (cyclin-dependent kinase 6) and acts as a negative regulator of normal cell proliferation [82]. For chromosome 10, the frequently involved regions are 10q21-22 and 10q24-qter [82].

In final stages of melanoma and in metastasis, gains on chromosome 6, 7, 8, and 1q were observed, and 1q chromosome gain is associated with 6p chromosome gain and 10q chromosome losses [82].

Loss of heterozygosity frequently concerns regions 1p, 9q, and 10q and is correlated with cell proliferation [82,83].

Chiu et al identified 249 copy number variations in circulating tumor cells (139 copy number gains and 110 copy number loss) found in more than 50% of the studied cases. A panel of five such CNV proved to have a negative prognostic impact (copy number gains: 1p35.1, 2q14.3, 14q32.33, and copy number loss: 14q32.11, 21q22.3) [84].

### 2.13. Hereditary Melanoma

Hereditary melanomas represent 5–12% of all melanomas [85]. Germline mutations are characterized by high penetrance and are associated with other cancers or other locations [85]. The mechanisms of oncogenesis include: activation of oncogenes, loss of tumor suppressor genes, and chromosomal instability. Most cancer predisposition syndromes are transmitted in an autosomal dominant manner [85,86,87]. Table 2 summarizes the genes and the syndromes with predisposition to cancer and risk for melanoma.

Lifetime melanoma risk is different in these syndromes: 60–90% in FAMMM or melanoma-pancreatic cancer syndrome, up to 80-fold in hereditary retinoblastoma, and 2000-fold in xeroderma pigmentosa [86].

## 3. Epigenomics in Melanoma and Implications for Biomarkers

Epigenetic changes relate to gene expression independent of changes in the DNA sequence that persist over several cell divisions [89].

Modifications of the epigenome are involved in cancer initiation, progression, and resistance to antitumor drugs [90]. Epigenetic marks are important as biomarkers for diagnosis, prognosis, predictive for disease recurrence, and therapeutic targets [91,92]. The epigenetic changes are reversible [93,94].

MicroRNAs, non-coding RNAs, histones modifications, and abnormal DNA methylations were associated with the stages of melanoma progression [95].

### 3.1. Histones Modifications

In histone modifications, 3 types of proteins are involved: histone writer proteins, eraser proteins, and reader proteins. The role of the first category is to add different chemical groups to histones through various chemical processes (acetylation, methylation, phosphorylation, ubiquitylation, glycosylation, ADP-ribosylation, carbonylation, and SUMOylation) [96]. Eraser proteins remove chemical groups. The reader proteins modulate gene expression by recruiting transcription factors or transcription repressors [90,97,98]. Chromatin remodelling occurs by modifying histones. These changes produce gene hyperexpression or gene inactivation [90]. The most important histone modifications are histone methylation and histone acetylation. 

#### 3.1.1. Methyltransferases and histone demethylases

Methylation of histones occurs most frequently at the level of lysine residues and/or arginine residues [96]. Lysine methylation consists of the addition of one to three methyl groups. The conversion from the unmethylated form to the methylated form is done under the action of methyltransferases (writer) and the reverse demethylation process is carried out by histone demethylases (eraser) [96,99]. Table 3 and Figure 1 summarize methyltransferases and histone demethylases, their action, and their place of action.

Histone methylation has two effects: silencing signatures and actively transcribed chromatin.

Silencing signatures are translated by the trimethylation of lysine 9, 27, and 36 at the N-terminal tail of histone H3 (H3-K9, H3-K27, H3-K36) and lysine 20 on histone H4 (H4-K20) [97]. H3-K9 trimethylation is a condition for establishing and maintaining a stable heterochromatin status [99]. Histone H3 acetylation at the level of 27 lysine residue (H3K27ac) is an important regulator of *MITF* (melanocyte inducing transcription factor) expression and is associated with an increased metastatic potential to melanoma cells [95,101].

Actively transcribed chromatin appears as a result of the trimethylation of lysine 4 and 79 of histone H3 (H3-K4, H3-K79) [99].

The most important methyltransferase with a role in melanoma is EZH2 (Histone-lysine N-methyltransferase EZH2) encoded by the *EZH2* gene (enhancer of zeste 2 polycomb repressive complex 2 subunit). EZH2 represents the catalytic subunit of Polycomb Repressive Complex 2 (PRC2). PRC2 has a role in (mono-, di-, or tri-) the methylation on H3 lysine 27 and in such a way in the silencing of target genes [100]. In total, 3% of melanomas are associated with activating mutations in *EZH2*, with a role in melanoma progression [97,102].

Acetylation is done exclusively at the level of lysine residues [94]. The balance of histone acetyl transferases and histone deacetylases determines the level and state of acetylation [97].

#### 3.1.2. Histone Deacetylases (HDACs) and Histone Acetyltransferases (Lysine Acetyltransferases—KATs)

Histone deacetylases (HDACs) are enzymes whose main function is removing acetyl groups from histones and secondary functional consequences appear on chromatin remodelling and gene expression. HDACs are classified into four classes: class I (HDAC1; HDAC2; HDAC3; HDAC8 – in nucleus); class II with subclass IIa (HDAC4; HDAC5; HDAC7; HDAC9 – in cytoplasm/nucleus) and subclass IIb (HDAC6; HDAC10 – in cytoplasm); class III (SIRT 1–7: cytoplasm/nucleus); and class IV (HDAC11 – in cytoplasm/nucleus) [96]. Class I, II, and IV are classical (Zn^2+^ dependent) and class III are NAD-dependent [103].

Several HDACs can act to deacetylate a certain histone in different forms of cancer: H3K9 (HDAC3), H3K14 (HDAC1, HDAC3), H3K56 (HDAC1, HDAC2), H4K5 (HDAC1, HDAC2, HDAC3), H4K8 (HDAC1, HDAC2), H4K12 (HDAC1, HDAC2, HDAC3), and H4K16 (SIRT1, SIRT2, HDAC1, HDAC2, HDAC3) [94].

Several acetylases can act to acetylate a lysine residue in different cancers: H3K9 (KAT2, KAT12), H3K14 (KAT2, KAT3A, KAT3B, KAT6, KAT10, KAT12), H3K18 (KAT2, KAT3, KAT12), H3K56 (KAT3B), and H4K16 (KAT5, KAT8) [94].

#### 3.1.3. Histone Variants

Histone variants contribute to epigenome plasticity [91].

The canonical histones are H2A, H2B, H3, and H4 [97,104]. The histone variants have different sequences and properties and can replace canonical histones. The effect is altered chromatin structures and gene transcription [97,104].

For the canonical H2A core histones, the expression is replication-dependent and the function is the core component of nucleosome. For non-canonical H2A core histones (H2A.X, H2A.Z, H2A.Z.2.1, H2A.Z.2.2, H2A.Bbd type 1, H2A.Bbd type 2 and H2A.Bbd type 3, macroH2A1.1, macroH2A1.2 and macroH2A2), the expression is replication-independent [105].

The histone variants macroH2A, H2A.Z, and H3.3 are important in melanoma [106]. The macroH2A has two regions: histone-like H2A domain and NHR (non-histone region); two isoforms (macroH2A1 and macroH2A2) and macroH2A1 have two spliced variants, macroH2A1.1 and macroH2A1.2. The role of macroH2A is a regulatory factor of transcription, cell differentiation, and reprogramming [106]. MacroH2A suppresses melanoma progression and its expression in melanoma is generally lost [97,107]. H2A.Z.2 and histone 3 variant H3.3 are highly expressed in melanoma. Overexpression of H2A.Z.2 produced activation of E2F (E2 factor of transcription) targets genes and consecutively promotes melanoma cell proliferation [97,108,109].

### 3.2. Long Non-Coding RNAs

The majority of the human genome transcript (90%) is not transcribed into proteins and has a role in the regulation of gene expression. Long non-coding RNA (lncRNAs) is represented by DNA sequences that has a length over 200 pb. It modifies the regulation of gene expression by transcription and translation regulation, chromatin changes, RNA changes through editing or splicing or degradation, and miRNA sequestration. At the cellular level, the modification of gene expression translates into the regulation of cell proliferation, differentiation, migration, and invasion [110].

Guo et al. synthesized the roles of lncRNAs in melanoma pathogenesis. Most lncRNA molecules are upregulated with a role in promoting melanoma pathogenesis: BASP1-AS1, SAMMSON, NCK1-AS1, LINC00470, LINC01291, MIR205HG, LINC00518, NEAT1, LHFPL3-AS1, TTN-AS1, LHFPL3-AS1, LINC00520, SRA, LINC00518, FOXD3-AS1, LNMAT1, SLNCR1, OIP5-AS1, CASC15, LncRNA-ATB, KCNQ1OT1, HOXD-AS1, FALEC, BANCR, CCAT1, PVT1, RHPN1-AS1, ANRIL, HEIH, TSLNC8, KCNQ1OT1, LINC01158, MALAT1, and ZEB1-AS1 [17].

lncRNA molecules that are downregulated with a suppressor role in melanoma pathogenesis have also been described: FUT8-AS1, TINCR, MEG3, LINC-PINT, DIRC3, ZNNT1, Linc00961, CPS1-IT1, CASC2, NKILA, CDR1as, GAS5, and LINC00459 [17].

In more than 90% of melanomas, an lncRNA SAMMSON, which plays an oncogenic role in association with MITF, is present [111].

Several lncRNAs were studied because of the high expression levels in melanoma patients, including SPRY4-IT1, MALAT-1, BANCR, UCA1, HOTAIR, and SNHG8. The levels of UCA1 and MALAT-1 were remarkably more elevated in patients with melanoma compared to healthy controls, and their levels were associated with the stage of the disease [112,113].

The expression of LINC02249 was found to be elevated in cutaneous melanoma. The high expression corresponds to poor OS and disease-specific survival, which is an independent prognostic factor [114].

### 3.3. MicroRNAs

Small ncRNAs includes miRNAs, piwi-interacting RNAs—piRNAs, and small nucleolar RNAs—snoRNAs. Their length is up to 200 nucleotides [115,116]. MiRNA (single stranded DNA molecules 18–25 nucleotides long) is involved in cancer by regulating oncogenes and tumour suppressor genes. The premature miRNAs are exported to the cytoplasm where mature miRNAs are formed. These molecules will bind to the target mRNA because of the complement of the bases. Mechanisms for regulating gene expression involve the degradation of the target mRNA or inhibition of translation into proteins [115,117].

Neagu et al. summarized the main miRNA in melanoma tissue and/or in circulation of patients with cutaneous melanoma: miR-214, miR-148a, miR-221, miR-16, miR-29c, miR-146a-5p, miR-205, pattern miR-142-5p, miR-150-5p, miR-342-3p, miR-155-5p, miR-146b-5p, miR-10b, miR-203, let-7a, and let-7b, miR-148, miR-155, miR-182, miR-200c, miR-211, miR-214,miR-221, miRNA-222, miR-106b, and 7 miRNAs (MELmiR-7). The following panel of oncomirs contains the miRNAs (miR-21, miR-10b, miR125b, miR-135b, miR-146a, miR-150, miR-155, miR-205, miR-211, miR-221, miR-222) that regulate tumour promoter genes associated with cutaneous melanoma (RAS/MAK, MITF, PI3K-AKT, P27Kip, NRAS, TYRP1, WEE1, LATS2). Some miRNAs function as tumour suppressors in melanoma: miR-16, miR-29c, miR-203, miR-205, miR-206, and miR-675. The genes targeted by these miRNAs are *DMMT3B*, *B7-H3*, and *MTDH* [118].

Huber et al. found a set of miRNAs (let-7e, miR-99b, miR-100, miR-125a, miR-125b, miR-146a, miR-146b, and miR-155) implicated in the transformation of monocytes into immunosuppressive MDSCs. MiRNAs such as miR-28 and miR-17-5p seem to interfere with PD-1 and PD-L1 expression at a post-transcriptional level, enabling resistance to immunotherapy [119].

Kanemaru et al. suggested that circulating miR-221-3p could be used as a melanoma biomarker, showing significantly distinct expressions between stage I/IV melanoma patients and healthy controls. Their work explained how miR-221-3p levels reduced after surgical removal of the primary tumor and increased upon disease recurrence, suggesting that circulating miRNA-221-3p could be a new tumor marker. High levels of miRNA-221 have been found in early-stage melanomas compared to healthy individuals. The levels of expression were also found to be proportionate with the stage of the disease [120].

### 3.4. Circular RNA

Circular RNAs (circRNAs) are endogenous RNA molecules with covalently looped structures [121,122]. There are several types of circRNA molecules according to their composition: exonic circRNAs (formed by exons), circular intronic RNAs (formed by introns), and exon-intron circRNAs (formed by both exons and introns) [123].

CircRNA intervenes in both normal physiological processes and pathological processes. CircRNAs modulate transcription, splicing, microRNA sponge, modulation of protein–protein interaction, and protein sponges [122].

Tang et al. summarized the circRNA involved in the inhibition of apoptosis, hyperproliferation, activation of invasion, migration, carbohydrate metabolism, and metastasis. Some of circRNA have an oncogenic function and activate several physiological processes in melanoma: circ_0084043 and circ-FOXM1 (proliferation, apoptosis, invasion). Other circRNA molecules have suppressive functions: circ_0023988 and circ_0030388 (proliferation, invasion, migration) [121].

Others circRNA molecules with low or increased expression have been described: circ_0016418 (up; promotes cell proliferation, migration, invasion, and epithelial to mesenchymal transition), ciRS-7 (CDR1as) (down; promotes cell invasion and metastasis, correlates with progression-free, overall survival and distinct therapeutic responses), and circ_0079593 (up; promotes cell proliferation, metastasis, glucose metabolism, inhibits apoptosis) [122,123,124,125]. CDR1as has been identified as a marker of progression in melanoma [126].

### 3.5. Abnormal DNA Methylation

DNA methylation is the biochemical process in which a methyl group is added to a cytosine or adenine at the 5-position of carbon where the DNA base thymine is located. The cytosine is converted to methylcytosine [127,128]. In total, 60% of gene promoters are associated with unmethylated CpG islands [128].

DNA methylation is a mark of suppression of gene expression. The effects are on cell differentiation and cell proliferation. Fu et al. summarized the genes hypermethylated in melanoma: *LINE-1*, *CLDN11*, *TERT*, *MGMT*, *KIT*, *TNF*, *MITF*, *RASSF6*, *RASSF10*, *GPX3*, *MMP-9*, *SYNPO2*, *CDKN1C*, *LXN*, *ASC/PYCARDC/PYCARD*, *Col11A1*, *SOCS1*, *Caspase 8*, *CDH1*, *MGMT*, *RAR-b2*, *CIITA-PIV*, *SOCS2*, *TNFRSF10C (DcR1/2)*, *TPM1*, *TIMP3*, *CDKN2A*, *DPPIV*, *FRZB*, *SOCS3*, *THBS1*, and *TM* [128].

Sigalotti et al. studied the importance of DNA methylation as a prognostic biomarker in stage III melanoma. They assessed the genome-wide methylation profiles from 45 patients. Based on global methylation, the cohort was split into a favorable group, with a median survival of 31.5 months, and an unfavorable group, with a median survival of 10.3 months, with a 5-year overall survival of 41.2 and 0%, respectively. The group identified a 17-gene methylation signature sufficient to distinguish the good prognosis group, characterized by low methylation density. Hypomethylation was a significant predictor of increased OS [129].

## 4. Conclusions

Cutaneous melanoma is the result of multiple genomic and epigenomic changes, some of them in close correlation. The negative prognosis of melanoma has raised interest in the discovery of prognostic and predictive biomarkers in order to improve life expectancy. For these reasons, the approach to the patient must be in relation to genetic and epigenetic changes. Prevention targeted the approaches of modifiable and non-modifiable risk factors. The existence of genetic and epigenetic biomarkers validated in various evolutionary stages would allow for a more precise molecular framing and a targeted approach to the changes present in the patient/group of similar patients. The general objective must be to avoid the metastatic stage.

## Figures and Tables

**Figure 1 ijms-24-00716-f001:**
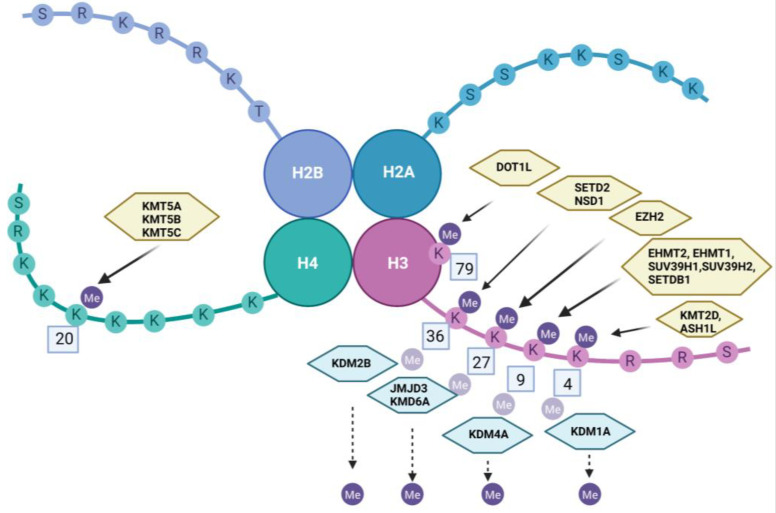
Methylation and demethylation of histones. Created with BioRender.com.

**Table 1 ijms-24-00716-t001:** Genes from MAPK pathways involved in melanoma pathogenesis [26,27].

Approved Gene Symbol	Approved Gene Name	Chromosomal Location	Protein	Frequency of Mutations in Melanoma (%)
*BRAF*	B-Raf proto-oncogene, serine/threonine kinase	7q34	Serine/threonine-protein kinase B-raf	53
*NRAS*	NRAS proto-oncogene, gtpase	1p13.2	GTPase NRas	32
*NF1*	Neurofibromin 1	17q11.2	Neurofibromin	19
*KIT*	KIT proto-oncogene, receptor tyrosine kinase	4q12	Mast/stem cell growth factor receptor Kit	8
*MAP2K1*	Mitogen-activated protein kinase kinase 1	15q22.31	Dual specificity mitogen-activated protein kinase kinase 1	7
*SPRED1*	Sprouty related EVH1 domain containing 1	15q14	Sprouty-related, EVH1 domain-containing protein 1	6
*GNA11*	G protein subunit alpha 11	19p13.3	Guanine nucleotide-binding protein subunit alpha-11	5
*KRAS*	KRAS proto-oncogene, gtpase	12p12.1	GTPase KRas	4
*MAP2K2*	Mitogen-activated protein kinase kinase 2	19p13.3	Dual specificity mitogen-activated protein kinase kinase 2	4
*MAPK1*	Mitogen-activated protein kinase 1	22q11.22	Mitogen-activated protein kinase 1	4
*GNAQ*	G protein subunit alpha q	9q21.2	Guanine nucleotide-binding protein G(q) subunit alpha	3
*MAPK3*	Mitogen-activated protein kinase 3	16p11.2	Mitogen-activated protein kinase 3	1.9
*HRAS*	Hras proto-oncogene, gtpase	11p15.5	GTPase HRas	1.9

**Table 2 ijms-24-00716-t002:** Syndromes with predisposition to cancer and risk for melanoma and their related genes/susceptibility locus [85,86,87,88].

Syndrome (Phenotype MIM Number)	Gene Symbol/Susceptibility Locus	Gene Name	Inheritance
Familial atypical multiple mole melanoma syndrome (FAMMM) (OMIM #155600)	*CDKN2A* (9p21.3) *CDK4* (12q14.1) *MC1R* (16q24.3) *XRCC3* (14q32.33) *MITF* (3p13) *TERT* (5p15.33) *POT1* (7q31.33) 1p36 1p22 20q11	cyclin dependent kinase inhibitor 2A cyclin dependent kinase 4 melanocortin 1 receptor X-ray repair cross complementing 3 melanocyte inducing transcription factor telomerase reverse transcriptase protection of telomeres 1	AD
Melanoma-pancreatic cancer syndrome (OMIM #606719)	*CDKN2A* (9p21.3)	cyclin dependent kinase inhibitor 2A	AD
Melanoma-astrocytoma syndrome (OMIM #155755)	*CDKN2A* (9p21.3)	cyclin dependent kinase inhibitor 2A	AD
Susceptibility to uveal melanoma 2 (OMIM #606661) Tumor predisposition syndrome-1 (OMIM# 614327)	*BAP1* (3p21.1) *BAP1* (3p21.1)	BRCA1-associated protein 1 BRCA1-associated protein 1	AD
Xeroderma pigmentosum XPA (OMIM#278700) XPB (OMIM#610651) XPC (OMIM#278720) XPD (OMIM#278730) XPE (OMIM#278740) XPF (OMIM#278760) XPG (OMIM#278780) XPV (OMIM#278750)	*XPA* (9q22.33) *ERCC3* (2q14.3) *XPC* (3p25.1) *ERCC2* (19q13.32) *DDB2* (11p11.2) *ERCC4* (16p13.12) *ERCC5* (13q33.1) *POLH* (6p21.1)	XPA, DNA damage recognition, and repair factor ERCC excision repair 3, TFIIH core complex helicase subunit XPC complex subunit, DNA damage recognition, and repair factor ERCC excision repair 2, TFIIH core complex helicase subunit damage specific DNA binding protein 2 ERCC excision repair 4, endonuclease catalytic subunit ERCC excision repair 5, endonuclease DNA polymerase eta	AR
Oculocutaneous albinism type 2 (OMIM#203200)	*OCA2* (15q12-q13.1)	OCA2 melanosomal transmembrane protein	AR
Hereditary Retinoblastoma (OMIM#180200)	*RB1* (13q14.2)	RB transcriptional corepressor 1	AD
Li-Fraumeni syndrome (OMIM#151623)	*TP53* (17p13.1)	tumor protein p53	AD
PTEN hamartoma tumor syndromes	*PTEN* (10q23.31)	phosphatase and tensin homolog	AD
Hereditary breast and ovary cancer syndrome	*BRCA1* (17q21.31) and *BRCA2* (13q13.1)	BRCA1 DNA repair associated BRCA2 DNA repair associated	AD

AD—autosomal dominant; AR—autosomal recessive.

**Table 3 ijms-24-00716-t003:** Main methyltransferases and histone demethylases and their actions [27,96,100].

	Location	Enzyme Symbol/Name	Gene Symbol/Name	Action
Methylation	H3K4	KMT2D (Histone-lysine N-methyltransferase 2D)	*KMT2D* (lysine methyltransferase 2D)	Catalyzes methyl group transfer from S-adenosyl-L-methionine to the epsilon-amino group of ‘Lys-4’ of histone H3 (H3K4)
		ASH1L (Histone-lysine N-methyltransferase ASH1L)	*ASH1L* (ASH1 like histone lysine methyltransferase)	Generates lysine 4-trimethylated histone h3k4me3
	H3K9	EHMT2 (G9a) (Histone-lysine N-methyltransferase EHMT2)	*EHMT2* (euchromatic histone lysine methyltransferase 2)	Specifically mono- and dimethylates ‘Lys-9’ of histone H3 (h3k9me1 and h3k9me2, respectively)
		EHMT1 (GLP) (Histone-lysine N-methyltransferase EHMT1)	*EHMT1* (euchromatic histone lysine methyltransferase 1)	Specifically mono- and dimethylates ‘Lys-9’ of histone H3 (h3k9me1 and h3k9me2
		SUV39H1 (Histone-lysine N-methyltransferase SUV39H1)	*SUV39H1* (SUV39H1 histone lysine methyltransferase)	Specifically trimethylates ‘Lys-9’ of histone H3 using monomethylated H3 ‘Lys-9’ as substrate
		SUV39H2 (Histone-lysine N-methyltransferase SUV39H2)	*SUV39H2* (SUV39H2 histone lysine methyltransferase)	Specifically trimethylates ‘Lys-9’ of histone H3 using monomethylated H3 ‘Lys-9’ as substrate
		SETDB1 (Histone-lysine N-methyltransferase SETDB1 )	*SETDB1* (SET domain bifurcated histone lysine methyltransferase 1)	Specifically trimethylates ‘Lys-9’ of histone H3
	H3K27	EZH2 (Histone-lysine N-methyltransferase EZH2 )	*EZH2* (enhancer of zeste 2 polycomb repressive complex 2 subunit)	Catalytic subunit of the PRC2/EED-EZH2 complex, which methylates ‘Lys-9’ (h3k9me) and ‘Lys-27’ (h3k27me) of histone H3
	H3K36	SETD2 (Histone-lysine N-methyltransferase SETD2)	*SETD2* (SET domain containing 2, histone lysine methyltransferase)	Specifically trimethylates ‘Lys-36’ of histone H3 (h3k36me3) using dimethylated ‘Lys-36’ (h3k36me2) as substrate
		NSD1 (Histone-lysine N-methyltransferase, H3 lysine-36 specific)	*NSD1* (nuclear receptor binding SET domain protein 1)	Dimethylates Lys-36 of histone H3 (h3k36me2)
	H3K79	DOT1L (Histone-lysine N-methyltransferase, H3 lysine-79 specific)	*DOT1L* (DOT1-like histone lysine methyltransferase)	Methylates ‘Lys-79’ of histone H3
	H4K20	KMT5A (N-lysine methyltransferase KMT5A)	*KMT5A* (lysine methyltransferase 5A)	Specifically monomethylates ‘Lys-20’ of histone H4 (h4k20me1)
		KMT5B (Histone-lysine N-methyltransferase KMT5B)	*KMT5B* (lysine methyltransferase 5B)	Specifically methylates monomethylated ‘Lys-20’ (h4k20me1) and dimethylated ‘Lys-20’ (h4k20me2) of histone H4 to produce respectively dimethylated ‘Lys-20’ (h4k20me2) and trimethylated ’Lys-20’ (h4k20me3)
		KMT5C (Histone-lysine N-methyltransferase KMT5C)	*KMT5C* (lysine methyltransferase 5C)	Specifically methylates monomethylated ‘Lys-20’ (h4k20me1) and dimethylated ‘Lys-20’ (h4k20me2) of histone H4 to produce respectively dimethylated ‘Lys-20’ (h4k20me2) and trimethylated ‘Lys-20’ (h4k20me3)
Demethylation	H3K4	KDM1A (Lysine-specific histone demethylase 1A)	*KDM1A* (lysine demethylase 1A)	Demethylate both ’Lys-4’ (h3k4me) and ‘Lys-9’ (h3k9me) of histone H3
		KDM5B (Lysine-specific demethylase 5B)	*KDM5B* (lysine demethylase 5B)	Demethylates ‘Lys-4’ of histone H3
	H3K9	KDM4A (Lysine-specific demethylase 4A)	*KDM4A* (lysine demethylase 4A)	Specifically demethylates ‘Lys-9’ and ‘Lys-36’ residues of histone H3
	H3K27	JMJD3 (Lysine-specific demethylase 6B)	*KDM6B* (lysine demethylase 6B)	Specifically demethylates ‘Lys-27’ of histone H3
		KMD6A (Lysine-specific demethylase 6A)	*KDM6A* (lysine demethylase 6A)	Specifically demethylates ‘Lys-27’ of histone H3
	H3K36	KDM2B (Lysine-specific demethylase 2B)	*KDM2B* (lysine demethylase 2B)	Demethylates ‘Lys-4’ and ‘Lys-36’ of histone H3

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
