# Peer review of "Genomics and Epigenomics in the Molecular Biology of Melanoma—A Prerequisite for Biomarkers Studies"

_ijms, 2022, doi:10.3390/ijms24010716_

Round 1

Reviewer 1 Report

This is a very well-written review article about the common mutations and lesions that occur in melanoma. Particularly useful are the sections on epigenetic changes and the other non-coding RNA changes that have been observed. Given the timely nature of this information and the thorough treatment provided by the authors, I highly recommend publication.

Author Response

This is a very well-written review article about the common mutations and lesions that occur in melanoma. Particularly useful are the sections on epigenetic changes and the other non-coding RNA changes that have been observed. Given the timely nature of this information and the thorough treatment provided by the authors, I highly recommend publication.

Answer: Thank you very much for your comments

Reviewer 2 Report

This is a well-written, comprehensive review regarding genomics and epigenomics of melanoma, which will be acceptable almost in the present form. However, I have found some minor points to be amended. I will list them up below:

1. (ll.116-117) These genomic abnormalities are usually not seen in nevi... & (ll.157-158) BRAF mutations are considered early events in the tumorigenesis of melanocytic tumor and are present in the majority of common neci. These sentences are contradictory to each other. Please adjust them.

2. (l.241, 280, ) References should be cited by reference numbers.

3. (ll.302-304) This sentence is grammatically incorrect and does not convey what the authors want to say.

4. (ll.299-328) These paragraphs should be headed as 2.9 INF-gamma and 2.10 TYRP1 and be independent of 2.8 TERT promoter mutations.

5. (ll.329-341) These paragraphs should be headed as 2.11 ctDNA and CTC and be independent of 2.8 TERT promoter mutations.

6. (ll.299-341) Instead of 4. and 5, these paragraphs may be headed as 2.9 Others.

7. (l.362) consists if: This may be “consists of.”

8. (ll.504-505) at the 5-position of the methyl group where the DNA base thymine is located: This may be “at the 5-position of carbon.”

9. (l.466) MDSCs should be spelled out.

Author Response

Answer: Thank you very much for your comments and constructive observations.

  1. (ll.116-117) These genomic abnormalities are usually not seen in nevi... & (ll.157-158) BRAF mutations are considered early events in the tumorigenesis of melanocytic tumor and are present in the majority of common neci. These sentences are contradictory to each other. Please adjust them.

Answer: We have corrected

  1. (l.241, 280, ) References should be cited by reference numbers.

Answer:We have corrected the mistake

  1. (ll.302-304) This sentence is grammatically incorrect and does not convey what the authors want to say.

Answer: We reorganized the text and removed the phrase

  1. (ll.299-328) These paragraphs should be headed as 2.9 INF-gamma and 2.10 TYRP1 and be independent of 2.8 TERT promoter mutations.
  2. (ll.329-341) These paragraphs should be headed as 2.11 ctDNA and CTC and be independent of 2.8 TERT promoter mutations.
  3. (ll.299-341) Instead of 4. and 5, these paragraphs may be headed as 2.9 Others.

Answer (4,5,6): We added

  1. (l.362) consists if: This may be “consists of.”

Answer: We replaced

  1. (ll.504-505) at the 5-position of the methyl group where the DNA base thymine is located: This may be “at the 5-position of carbon.”

Answer: We replaced

  1. (l.466) MDSCs should be spelled out.

Answer:We added

Reviewer 3 Report

Zob and colleagues review some of the genomics and epigenomics of cutaneous melanoma. 

Although much is addressed - the manuscript starts strong - this review is far from complete and is mainly addressing rare variants in driver genes and summarizes only part of the genetics of primary melanoma. For instance there is genetic predisposition (common variants) are not  addressed extensively as are copy number variations, tumor evolution, the relationship to metastasis and survival. Much of the epigenomics feels more like a summery of genes and micro RNA's.  

Author Response

Zob and colleagues review some of the genomics and epigenomics of cutaneous melanoma. 

Although much is addressed - the manuscript starts strong - this review is far from complete and is mainly addressing rare variants in driver genes and summarizes only part of the genetics of primary melanoma. For instance there is genetic predisposition (common variants) are not  addressed extensively as are copy number variations, tumor evolution, the relationship to metastasis and survival. Much of the epigenomics feels more like a summery of genes and micro RNA's.  

Thank you very much for your comments and constructive observations.

Answer: We reorganized the text and we added more information concerning the common variants, the genes and the syndromes with predisposition to cancer and risk of melanoma. We added also a figure in order to better explain the epigenomics.   

Reviewer 4 Report

The authors described the problem of genomics and epigenomics in the molecular biology of Melanoma in a very interesting and comprehensive way.
The study is very detailed and is based on many literature items, including the latest ones.

However, I have a few suggestions:

- it would be advisable to emphasize the future perspectives and the use of the information described in the article

- the introduction of some diagram or graphics to the article would diversify the work and better explain the issues.

Nevertheless, the article is good and changes would improve its value, which allows for publication in the journal.

Author Response

The authors described the problem of genomics and epigenomics in the molecular biology of Melanoma in a very interesting and comprehensive way.
The study is very detailed and is based on many literature items, including the latest ones.

However, I have a few suggestions:

- it would be advisable to emphasize the future perspectives and the use of the information described in the article

- the introduction of some diagram or graphics to the article would diversify the work and better explain the issues.

Nevertheless, the article is good and changes would improve its value, which allows for publication in the journal.

Answer: We reorganized the text and we added more information concerning the common variants, the genes and the syndromes with predisposition to cancer and risk of melanoma. We add also a figure in order to better explain the epigenomics. We added some future perspectives in conclusions 

Round 2

Reviewer 3 Report

Authors added a nice figure addressing epigenomics and restructured the manuscript and added a syndromic cancer predisposition table. A final version (without track changes) would have been nice to review. Perhaps because of this, I could not find mentioning of common variants (see for instance Landi et al Nature genetics 52, 494–504 (2020)) nor CNV (see for instance TCGA, Cell 161, 1681–1696, 2015, Bastian et al, Cancer Research 58, 10, 2170-2175 (1998). Especially in the last sections, the manuscript reads more like a summary and does not go into much detail (at least not as extensively for each subsection). The relationship between the biomarkers to each other, survival and treatment is not addressed for all “biomarkers”. The manuscript would benefit substantially if authors would provide a complete overview of all relevant genomic changes, relating these to regions, genes, risk, survival and treatment as well as with biology. 

Author Response

Thank you very much for help us to improve the manuscript.

We added two subsections with common variants and with copy number variations in melanoma. We also completed the epigenomic section.

Round 3

Reviewer 3 Report

I have no further suggestions